# Electrochemical Disinfection of Dental Implants Experimentally Contaminated with Microorganisms as a Model for Periimplantitis

**DOI:** 10.3390/jcm9020475

**Published:** 2020-02-09

**Authors:** Maximilian Koch, Maximilian Göltz, Meng Xiangjun, Matthias Karl, Stefan Rosiwal, Andreas Burkovski

**Affiliations:** 1Microbiology Division, Department of Biology, University of Erlangen-Nuremberg, 91058 Erlangen, Germany; Maximilian.G.F.Koch@gmx.de (M.K.); mengxiangjun1992@gmail.com (M.X.); Andreas.burkovski@fau.de (A.B.); 2Division of ultra-hard coatings, Department of Material Sciences, University of Erlangen-Nuremberg, 91058 Erlangen, Germany; maximilian.goeltz@fau.de; 3Department of Prosthodontics, Saarland University, 66421 Homburg/Saar, Germany; 4Chair of Materials Science and Engineering for Metals, Department of Material Sciences, University of Erlangen-Nuremberg, 91058 Erlangen, Germany; Stefan.Rosiwal@fau.de

**Keywords:** air abrasion, antimicrobial treatment, biofilm, boron-doped diamond, mechanical debridement, reactive oxygen species

## Abstract

Despite several methods having been described for disinfecting implants affected by periimplantitis, none of these are universally effective and may even alter surfaces and mechanical properties of implants. Boron-doped diamond (BDD) electrodes were fabricated from niobium wires and assembled as a single instrument for implant cleaning. Chemo-mechanical debridement and air abrasion were used as control methods. Different mono-species biofilms, formed by bacteria and yeasts, were allowed to develop in rich medium at 37 °C for three days. In addition, natural multi-species biofilms were treated. Implants were placed in silicone, polyurethane foam and bovine ribs for simulating different clinical conditions. Following treatment, the implants were rolled on blood agar plates, which were subsequently incubated at 37 °C and microbial growth was analyzed. Complete electrochemical disinfection of implant surfaces was achieved with a maximum treatment time of 20 min for *Candida albicans*, *Candida dubliniensis*, *Enterococcus faecalis*, *Roseomonas mucosa*, *Staphylococcus epidermidis* and *Streptococcus sanguinis*, while in case of spore-forming *Bacillus pumilus* and *Bacillus subtilis*, a number of colonies appeared after BDD electrode treatment indicating an incomplete disinfection. Independent of the species tested, complete disinfection was never achieved when conventional techniques were used. During treatment with BDD electrodes, only minor changes in temperature and pH value were observed. The instrument used here requires optimization so that higher charge quantities can be applied in shorter treatment times.

## 1. Introduction

While a consistent definition of periimplantitis [1] as well as diagnostic criteria defining this disease are still missing [2,3], periimplantitis has received tremendous attention during the past years [3]. In addition, considerable heterogeneity exists among reports on the prevalence of periimplantitis [3]. A systematic review and meta-analysis conducted by Dreyer and coworkers found prevalence on implant level between 1.1% and 85.0% [2], while Rakic and coworkers found prevalence of 12.8% at implant level [4]. Irrespective of the exact numbers, periimplantitis remains being a complication, threatening long-term implant survival [5,6].

According to a recent literature review, periimplantitis has a multifactorial etiology [3] for which several risk factors [7,8] have been described. These include patient-specific factors such as genetic disorders, smoking [1] and periodontal disease [9,10], cement or impression material remnants in the periimplant sulcus [11,12], bacterial contamination of the implant components [13], technical issues and implant surface characteristics [5,10,14].

Several attempts already have been made aimed at characterizing periimplant lesions as well as differences between teeth and implants with respect to pathologic processes [15,16]. While the exact patho-mechanisms seem not to be fully understood, it appears to be consensus that bacterial biofilms on dental implants [13,17] can cause an inflammatory reaction [12,18,19], resulting in loss of periimplant bone [20,21]. However, it has also been pointed out that marginal bone loss around dental implants may result from the bone’s response to surgical trauma and implant loading which must be differentiated from periimplantitis [22].

Most treatment strategies depend on the clinical situation considering probing depth, suppuration, periodontal indices such as bleeding on probing and plaque index as well as radiographic bone loss [3,23,24,25] as decisive factors. Treatment may then range from implant debridement, resective and reconstructive surgery [23] to implant removal [26]. According to several authors, treatment of periimplantitis is considered as having an unpredictable outcome [6,20] in particular when evaluating the effectiveness of regenerative treatment [23,27].

A broad variety of techniques for the disinfection of dental implants has been described in the literature [26]. These include mechanical instrumentation [28,29], chemical and antimicrobial agents [30], treatment with local or systemic antibiotics [17,29,31], laser application [32], photodynamic therapy [18,33,34], cold plasma treatment [35,36] and air abrasion [37,38,39,40]. In many instances, only combinations of different disinfection techniques have been shown to be effective [36,40,41,42], with mechanical debridement bearing the risk of implant surface alterations [43]. A novel and just recently published approach employs electrochemical principles for in situ removal of biofilm from textured dental implant surfaces using the implant itself as an electrode [44].

In this preliminary proof of principle study, we tested the application of boron-doped diamond (BDD) electrodes [45,46,47] for the electrochemical disinfection of dental implants colonized by biofilm-forming microorganisms. The working principle of these electrodes is based on the electrolytic dissociation of water, which theoretically produces hydrogen at the negative pole (cathode) and oxygen at the positive pole (anode). Due to the properties of the electrode material used, a higher voltage than theoretically needed is required (overpotential). When diamond electrodes are being used, an overpotential of 2.8 V is required for anodic oxygen production while the desired disinfective OH radicals are generated already at 2.5 V.

## 2. Materials and Methods

### 2.1. Preparation of Electrodes

Diamond coating with boron doping of thin niobium wires (200 µm in diameter) was performed in a Hot-Filament Chemical Vapor Deposition machine at approx. 800 °C in a methane-hydrogen-trimethylborate gas atmosphere at 2 mbar for 6 h. The functionality and the manageability of the diamond coatings where verified using bending tests. Optimal properties with respect to stiffness and surface roughness were obtained when the niobium wires were sandblasted (air pressure 4 bar) with silicon-carbide particles approximately 46 µm in size prior to BDD coating. A special wire coating setup was developed to minimize deformation of the wires during diamond coating. With this approach, a reproducible thickness of the dense diamond coating of approximately 2 µm could be achieved (Figure 1). Two of these electrodes were combined with electrical insulating media to form a probe-like instrument with clinically applicable dimensions (Figure 2). This probe was connected to an external electric power supply allowing for adjusting voltage and treatment time.

### 2.2. Treatment of Experimentally Contaminated Implants as Periimplantitis Model

Forty-five commercially available dental implants with a medium rough surface (Straumann Bone Level Tapered 4.1 x 12 mm RC; REF: 021.5512; LOT: RP027) were exposed to different microbes for three days at 37 °C in rich medium [Brain Heart Infusion (BHI); Oxoid, Wesel, Germany] to allow biofilm formation on external and internal surfaces (Note: Implants were reused following sterilization in an autoclave in order to increase sample size; please cf. Appendix A). The microorganisms applied included yeasts (*Candida albicans*, *Candida dubliniensis*), Gram-negative (*Roseomonas mucosa*) and Gram-positive bacteria (*Enterococcus faecalis*, *Staphylococcus epidermidis*, *Streptococcus sanguinis*), including spore-forming bacteria (*Bacillus pumilus*, *Bacillus subtilis*) (Table 1). The microorganisms were chosen based on their robustness and occurrence in cases of infected root canals [31,32] as well as in cases of peri-implantitis [33,34]. Often found members of the genera *Prevotella* and *Treponema* were not included in the study, since these anaerobic and microaerophilic bacteria are highly oxygen-sensitive and less resistant against reactive oxygen species and even atmospheric oxygen concentrations. For control of biofilm formation, staining and quantitative analysis was carried out as described previously [48].

The implants were either placed in (i) elastic silicone, stable against temperature, acid, base and oxidants and bacterial colonization (Bindulin, Fürth, Germany), mimicking periimplant soft tissue, polyurethane foam blocks (Cellular Rigid polyurethane foam 20pcf, Sawbones Europe AB, Malmö, Sweden) mimicking type IV alveolar bone (Figure 3) according to the Lekholm and Zarb classification [51] or in bovine ribs. In the latter cases, osteotomies were created in preformed saucer-shaped defects applying the regular surgical protocol [39]. The defects simulated circular bone resorption under maintenance of the buccal and oral compacta resembling class Ie defects according to a clinical classification system [52]. These specimens were placed in containers filled with phosphate-buffered saline (137 mM NaCl, 2.7 mM KCl, 10 mM Na_2_HPO_4_, 2 mM KH_2_PO_4_, pH 7.4) and treated applying the following methods: (i) Mechanical debridement (stainless steel curettes (EXD11/12, HuFriedy, Chicago, IL, USA), polishing of accessible implant surfaces) and irrigation with chlorhexidine (Chlorhexamed FORTE ethanol-free 0.2%, GlaxoSmithKline Consumer Healthcare GmbH & Co. KG, Munich, Germany) for a total of 5 min. (ii) Air abrasion (AIRFLOW PLUS, EMS ElectroMedicalSystems GmbH, Munich, Germany) and irrigation with chlorhexidine (Chlorhexamed FORTE ethanol-free 0.2%, GlaxoSmithKline Consumer Healthcare GmbH & Co. KG) for a total of 5 min. (iii) Electrochemical disinfection using the electrode configuration described above applying different treatment times. For every species investigated, at least 3 biological replicates were tested for each treatment procedure.

After cleaning and disinfection, the implants were rolled five to seven times on Columbia Blood Agar plates (Oxoid, Wesel, Germany), which were subsequently incubated at 37 °C for one day. Bacterial growth was monitored and rated using an evaluation scheme adapted from a monitoring scheme of catheter infections [53]. To this end, each lane of the roll-out was rated from 0 (no growth) to 3 (strong growth) using a master sample as reference (Figure 4). All experiments were carried out in independent replicates (*n* = 3 biological replicates).

### 2.3. Statistical Analysis

The ratings obtained for mechanical debridement, air abrasion and 5 min BDD electrode treatment of implants (*n* = 3) covered by *E. faecalis*, *C. dubliniensis* and multi-species biofilm were subject to comparative statistical analysis. To this end, the sums of the individual ratings were used. In view of the small sample sizes, the integer valued results cannot be assumed to be normally distributed. Therefore, the parameter-free Kruskal Wallis rank sum test (R, version 3.6.2, The R Foundation for Statistical Computing, Vienna, Austria; www.R-project.org) was applied to compare the different treatments. The level of significance was set at α = 0.05.

## 3. Results

### 3.1. Biofilm Formation on Experimetally Contaminated Implants

It was the aim of this study, to investigate the elimination of biofilms on implants using BDD electrodes as new electrochemical treatment method for periimplantitis. As a prerequisite of our experiments, implants were incubated in rich medium with distinct microbial species for several days. Subsequently, biofilm formation was tested by staining with crystal violet solution. In all cases, biofilm formation on implants was observed; however, in a species-specific amount. The strongest mono-species biofilm producers were *B. subtilis* and *C. dubliniensis*, moderate amounts of biofilm were produced by *B. pumilus*, *R. mucosa* and *S. sanguinis*, while the poorest colonization of implants tested here was observed for *C. albicans*, *E. faecalis* and *S. epidermidis* (Figure 5).

### 3.2. Removal of Biofilm from Implants Contaminated with C. dubliniensis

BDD application to implants placed in both, silicone and polyurethane foam for 10 min at constant 6 V was at least as effective as the control treatments i.e. mechanical debridement and air abrasion. Increasing the treatment times of BDD electrodes led to even better results (Figure 6). 

### 3.3. Removal of Biofilm from Implants Contaminated with E. faecalis

Mechanical debridement and air abrasion were not suitable for complete elimination of *E. faecalis* biofilm and disinfection of the implants (Figure 7). In contrast, using BDD electrode treatment, full removal of biofilm and complete disinfection of implants was achieved within 5 to 10 min depending on the model system used (Figure 7).

### 3.4. Removal of Multispecies Biofilms from Implants

When bovine ribs were used to insert implants, a multi-species colonization was observed, despite the fact that only *E. faecalis* was used for pre-incubation and colonization. Obviously, the tested bone material was already strongly contaminated with a number of different microorganisms and, consequently, a natural multi-species biofilm developed in the prepared bone. Treatment of these implants showed an inferior disinfection success. This result may be explained by the putative presence of spore-forming bacteria in the uncharacterized natural multi-species biofilm and the fact that bone debris, which was not removed prior to rolling the specimens on the blood agar plates, was sticking on the implant surface. However, compared to mechanical debridement and air abrasion, BDD treatment still performed best (Figure 8).

### 3.5. Statistical Comparison Between Treatment Methods

The mean values of the ratings as well as the results of the Kruskal Wallis rank sum tests for comparing the three different treatment modalities are given in Table 2. In no instance, a statistically significant difference could be observed between the disinfection techniques applied.

### 3.6. Time-Dependent Removal of Biofilm from Implants Contaminated with Different Microorganisms

In addition to the microorganisms mentioned above and multi-species biofilm, the effect of BDD electrode treatment on mono-species biofilms formed by a number of different bacteria and yeasts was tested. The fastest elimination (within 10 min) was achieved for *E. faecalis*, *R. mucosa*, *S. sanguinis* and *C. dubliniensis*. A slightly longer treatment time of 20 min was necessary for the disinfection of implants colonized by *C. albicans* and *S. epidermidis*. In case of spore-forming *B. pumilus* and *B. subtilis*, a significant reduction of growth but no complete disinfection was reached under the experimental conditions applied due to the formation of highly resistant spores (6 V, 5–22 mA) (Table 3).

### 3.7. Influence of BDD Electrode Treatment on Temperature and pH

Successful treatment of periimplantitis must not only rely on a reproducible disinfection protocol, but also on the absence of negative side effects. Therefore, changes in physico-chemical parameters depending on BDD treatment were tested. After treatment times of 25 min, changes in temperature in the range of 2 °C and changes in pH of 1 unit were recorded in a controlled setting with a reaction tube filled with 4 ml of phosphate-buffered saline and the respective probes mounted in the direct surrounding of the electrodes (Figure 9).

## 4. Discussion

Despite using a simplistic prototype, it was shown that electrochemical disinfection of contaminated implant surfaces was possible when BDD electrodes based on niobium wires were used. In contrast to mechanical debridement damaging the implant surface and air powder abrasion leaving powder remnants on the implant surface, no alterations of the implant surfaces were identified with the use of BDD electrodes [54]. As expected, the different microbes tested in this study showed varying levels of sensitivity and hence required varying amounts of treatment time until complete disinfection was achieved. The worst performer in this respect were, besides an uncharacterized multi-species biofilm, *B. pumilus* and *B. subtilis*, both forming highly resistant spores under the experimental conditions applied, while *E. faecalis*, *R. mucosa* and *S. sanguinis* were safely eliminated with a maximum treatment time of 10 min. With the non-optimized setup used here, these treatment times would be too long for clinical application. The critical aspect, however, is not the treatment time but the charge quantity applied. Future developments will hence include modified electrodes with increased surfaces.

Minor temperature and pH-value changes were observed after applying the electrodes for approximately 25 min and hence would not be expected in a clinical setting where clearly shorter treatment times are needed. Even if these changes would occur, negative effects on the patient cannot be expected.

It had been anticipated that disinfection would become more difficult when implants are placed in bone surrogate materials or cadaver bone as compared to having full access to all critical surfaces. While not expressible as a quantitative effect, this correlation was observed for the control treatments of chemo-mechanical debridement and air abrasion, respectively. In contrast, when electrochemical disinfection was applied, restricted access to implant surfaces obviously had less effect.

Recently, an electro-chemical approach for implant debridement and periimplantitis treatment was introduced utilizing the implant itself as an electrode (cathode) while a wire, not contacting the infected surface, was used as another electrode (anode) [44]. By this approach, not using BDD-coated electrodes, biofilm appears to be primarily removed by bubbles of electrochemically generated hydrogen at the cathodic implant surface. This theoretically bears the risk of spreading the biofilm into the surrounding tissue and may also generate corrosion problems on the implant surface by hindering the self-passivation of titanium after damaging the native oxide layer by a mechanical load. Furthermore, the hydrogen uptake of the titanium implant can lower the mechanical strength of the material by hydrogen embrittlement [55].

Given that the experiment at hand constituted the first application of BDD electrodes for biofilm removal from dental implant surfaces, a number of limitations have to be taken into account when interpreting the findings presented. A wide variety of bacteria and yeasts have been shown to be present in dental biofilms, while in the current experiment mostly mono-species biofilms were considered. Although different bacteria may form a synergistic biofilm, their resistance to disinfecting measures is not influenced by other species. However, with the spore-forming species used here, a worst-case scenario has already been tested. It has been shown that not only the clinical situation per se but also the defect morphology impacts the result of periimplantitis treatment [52]. As such, the model situations chosen here clearly present simplifications of reality, which were needed due to the prototypical stage of the BDD electrode setup. Due to the exploratory nature of this pilot investigation, comparative statistical analysis among the treatment modalities applied was limited to a subset of experiments where implants had been treated for exactly 5 min using all three disinfection methods. Future studies are under way with much greater sample size aimed at quantitatively comparing different treatment modalities.

Taking into account that numerous developmental steps including preclinical and clinical studies will be required prior to clinical application of an instrument based on a BDD electrode array, a probe-like instrument with a permeable cover can already be envisaged. Not requiring superstructure removal as well as the universal applicability also in periodontal and endodontic treatment would be major advantages of such an instrument.

## 5. Patents

Stefan Rosiwal, Andreas Burkovski, Maximilian Göltz and Matthias Karl have a filed a patent for the disinfection method described in this report.

## Figures and Tables

**Figure 1 jcm-09-00475-f001:**
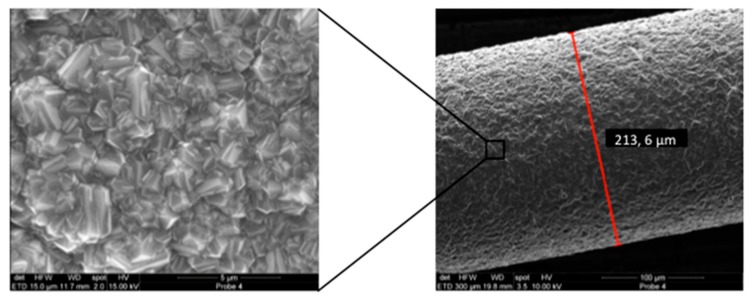
Scanning electron microscopy (SEM) image of a niobium wire after boron doped diamond (BDD) coating (Right: Overview; Left: Close up view of the area indicated by black rectangle). The coating layer has an approximate thickness of 2 µm and shows no delamination.

**Figure 2 jcm-09-00475-f002:**
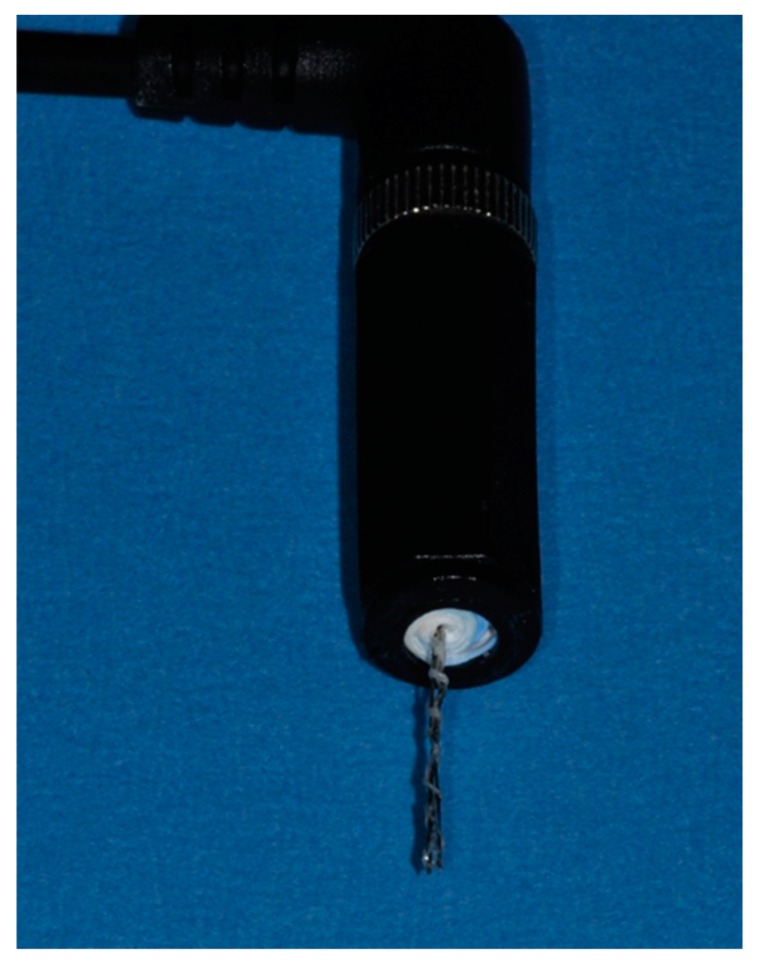
Disinfection apparatus consisting of an electrode array with intermediate insulating material.

**Figure 3 jcm-09-00475-f003:**
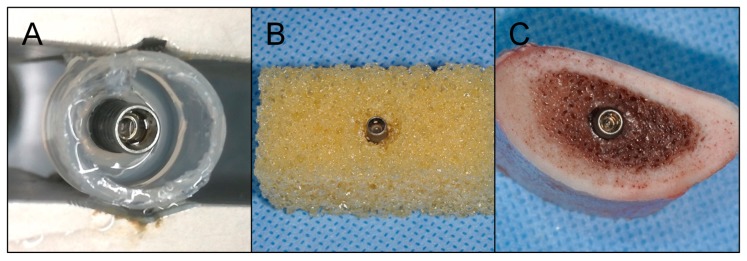
Placement of dental implants in model substrates. (**A**) Silicone simulating periimplant soft tissue, (**B**) polyurethane foam mimicking type IV alveolar bone with a circumferential defect and (**C**) bovine ribs.

**Figure 4 jcm-09-00475-f004:**
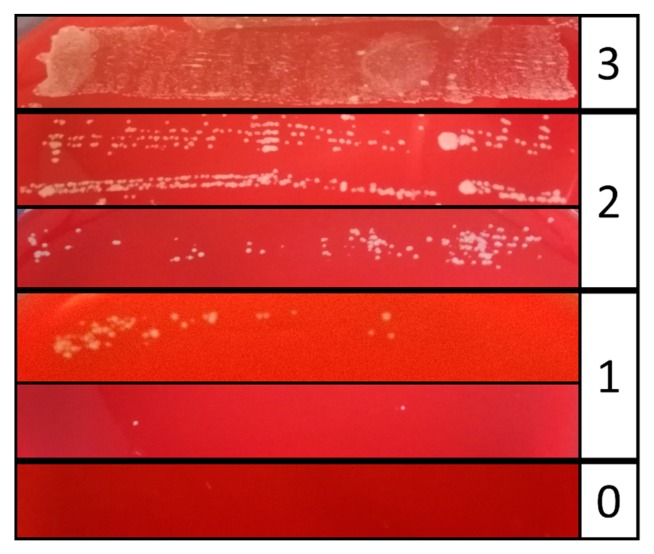
Evaluation scheme for microbial growth. Treated implants were rolled on Columbia Blood Agar plates and incubated at 37 °C. Growth of the individual roll-outs was scored from 0 = no bacterial growth, 1 = minor growth, 2 = major growth, 3 = strong growth.

**Figure 5 jcm-09-00475-f005:**
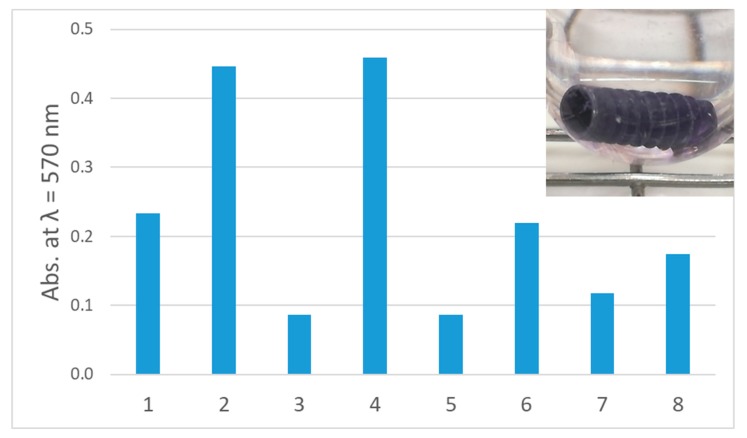
Biofilm formation on implants. Quantitative analysis of biofilm formation: (1) *B. pumilus*, (2) *B. subtilis*, (3) *C. albicans*, (4) *C. dubliniensis*, (5) *E. faecalis*, (6) *R. mucosa*, (7) *S. epidermidis*, (8) *S. sanguinis*. Inset: Crystal violet staining of *R. mucosa* biofilm.

**Figure 6 jcm-09-00475-f006:**
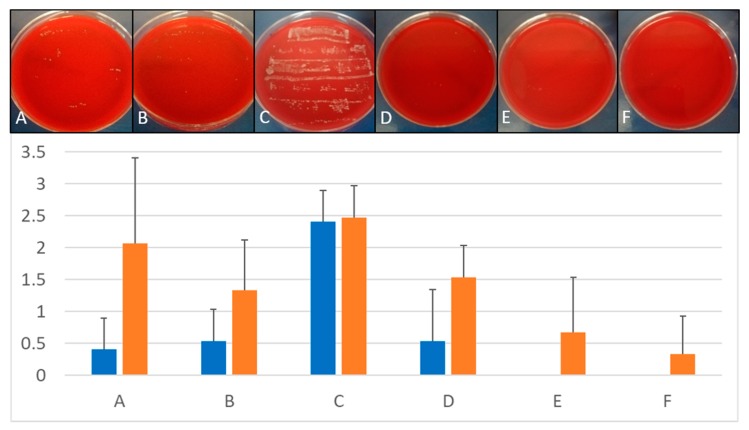
Treatment of *C. dubliniensis* biofilms. Upper panel: Comparison of growth on Columbia Blood Agar plates after different treatment of implants infected with *C. dubliniensis* and placed in silicone. Lower panel: Quantitative comparison of growth on implants inserted in silicone (blue bars) and polyurethane foam (orange) depending on treatment protocols. (**A**) Mechanical debridement, (**B**) air abrasion, (**C**) BDD treatment for 0 min, (**D**) BDD treatment for 5 min, (**E**) BDD treatment for 10 min, (**F**) BDD treatment for 15 min (*n* = 3 for each methods and time point). Growth of implant roll-outs was rated from 0 (no growth) to 3 (strong growth). Columns represent the mean of 3 independent biological replicated ± standard deviation (SD).

**Figure 7 jcm-09-00475-f007:**
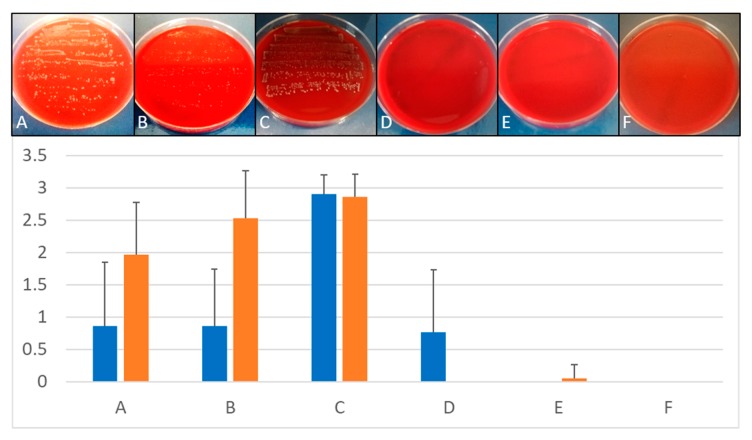
Treatment of *E. faecalis* biofilms. Upper panel: Comparison of growth on Columbia Blood Agar plates after different treatment of implants infected with *E. faecalis* and placed in silicone. Lower panel: Quantitative comparison of growth on implants inserted in silicone (blue bars) and polyurethane foam (orange bars) depending on treatment protocols. (**A**) Mechanical debridement, (**B**) air abrasion, (**C**) BDD treatment for 0 min, (**D**) BDD treatment for 5 min, (**E**) BDD treatment for 10 min, (**F**) BDD treatment for 15 min (*n* = 3 for each methods and time point). Growth of implant roll-outs was rated from 0 (no growth) to 3 (strong growth). Columns represent the mean of 3 independent biological replicated ± standard deviation (SD).

**Figure 8 jcm-09-00475-f008:**
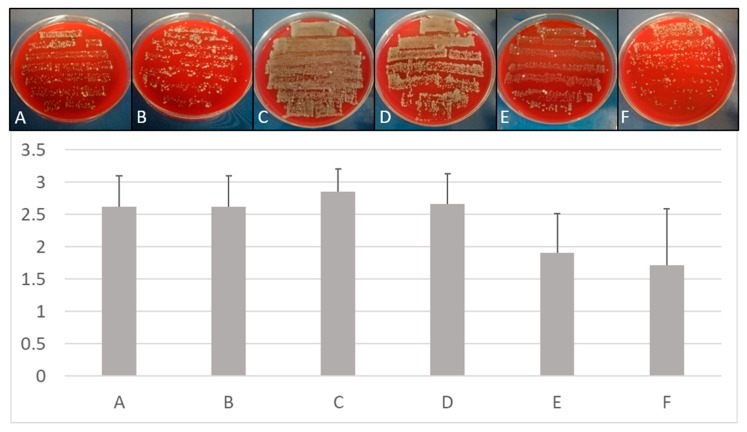
Treatment of multi-species biofilms. Upper panel: Comparison of growth on Columbia Blood Agar plates after different treatment of implants placed in bovine cadaver bone. Lower panel: Quantitative comparison of growth on implants inserted in ribs depending on treatment protocols. (**A**) Mechanical debridement, (**B**) air abrasion, (**C**) BDD treatment for 0 min, (**D**) BDD treatment for 5 min, (**E**) BDD treatment for 10 min, (**F**) BDD treatment for 15 min (*n* = 3 for each methods and time point). Growth of implant roll-outs was rated from 0 (no growth) to 3 (strong growth). Columns represent the mean of 3 independent biological replicated ± standard deviation (SD).

**Figure 9 jcm-09-00475-f009:**
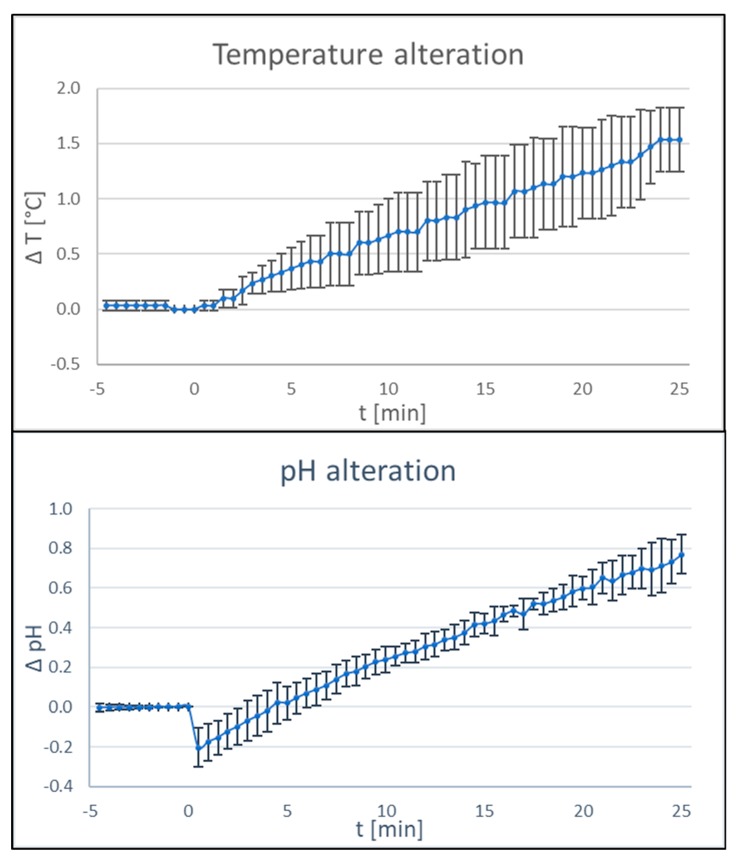
Influence of BDD electrode treatment on physico-chemical parameters. Changes in pH (upper panel) and temperature (lower panel) during application of the BDD electrode in phosphate-buffered saline was tested. For each parameter, three independent experiments were carried out (*n* = 3, current 15–20 mA; voltage 6 V).

**Table 1 jcm-09-00475-t001:** Microorganisms used in this study.

Species	Description	Reference/Source
**Bacteria**		
*Bacillus pumilus*	Gram-positive isolate from infected root canal, spore former	[49]
*Bacillus subtilis*	Strain 168, Gram-positive spore former	Bacillus Genetic Stock Centre (Columbus, OH, USA)
*Enterococcus faecalis*	DSM 20478, type strain, Gram-positive	German Type Culture Collection, DSMZ, Braunschweig, Germany
*Roseomonas mucosa*	Gram-negative isolate from infected root canal	[50]
*Staphylococcus epidermidis*	DSM 20044, type strain, Gram-positive	German Type Culture Collection, DSMZ, Braunschweig, Germany
*Streptococcus sanguinis*	DSM 20567, type strain, Gram-positive	German Type Culture Collection, DSMZ, Braunschweig, Germany
**Yeasts**		
*Candida albicans*	Strain SC5314	Laboratory stock
*Candida dubliniensis*	Isolate from infected root canal	[50]

**Table 2 jcm-09-00475-t002:** Mean values and standard deviations of ratings recorded for implants contaminated with *E. faecalis*, *C. dubliniensis* (implants placed in silicone or polyurethane foam) and multispecies biofilm (implants placed in bovine rib) following 5 min of treatment with mechanical debridement, air abrasion and BDD electrode (*n* = 3 per microorganism and treatment method). Comparative statistics did not reveal any significant differences among the three modalities (*p* > 0.05).

Microbial Species	Mechanical Debridement		Air Abrasion		Electrochemical Disinfection		Kruskal Wallis Chi-squared	df	*p*-Value
	Mean	SD	Mean	SD	Mean	SD			
*E. faecalis*, polyurethane	1.90	0.83	2.50	0.76	0.00	0.00	5.804	2	0.055
*E. faecalis*, silicone	0.86	1.01	0.86	0.91	0.76	1.00	0.067	2	0.967
*C. dubliniensis*, polyurethane	2.07	1.39	1.33	0.82	1.53	0.52	0.605	2	0.739
*C. dubliniensis*, silicone	0.40	0.51	0.53	0.52	0.53	0.83	0.487	2	0.784
Multi-species biofilm, bovine rib	2.62	0.50	2.62	0.50	2.67	0.48	0.318	2	0.853

**Table 3 jcm-09-00475-t003:** Time for disinfection of implants placed in silicone depending on biofilm-forming species. All experiments were carried out in three independent biological replicates (*n* = 3) and the longest necessary treatment time is given. Implants were pre-incubated with the respective microorganisms to allow biofilm formation for three to five days.

Species	Maximum Time
**Bacteria**	
*B. pumilus*	Incomplete disinfection within 60 min
*B. subtilis*	Incomplete disinfection within 60 min
*E. faecalis*	10 min
*R. mucosa*	10 min
*S. sanguinis*	10 min
*S. epidermidis*	20 min
**Yeasts**	
*C. albicans*	20 min
*C. dubliniensis*	10 min
**Mixture of microorganisms**	
Multi-species natural biofilm developed on bovine ribs *	Incomplete disinfection within 15 min

* Implants were pre-incubated with *E. faecalis*, but bacteria were overgrown by a mixture of microorganisms already present on the cadaver bone.

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
