# Peer review of "Electrochemical Disinfection of Dental Implants Experimentally Contaminated with Microorganisms as a Model for Periimplantitis"

_jcm, 2020, doi:10.3390/jcm9020475_

Round 1

Reviewer 1 Report

The authors present a promising technology for disinfection of contaminated dental implants. The methods are valid and the results are well-presented. On the other hand, it should be taken into account that this is a very basic research and clinical studies should be performed in order to test the applicability of this technology in everyday clinical practice. 

Author Response

Thank you very much for your review.

In addition to the multiple statements already made in the text with respect to the pilot character of the study / technology, we have further emphasized this aspect in the last paragraph of the Discussion where it now reads:

"Taking into account that numerous developmental steps including preclinical and clinical studies will be required prior to clinical application of an instrument based on a BDD electrode array, a probe-like instrument with a permeable cover can already be envisaged"

Reviewer 2 Report

The revised version of the article “Electrochemical Disinfection of Dental Implants Experimentally Contaminated with Microorganisms as a Model for Periimplantitis”  showed some improvements.  However, because the improvements are not properly listed, it is hard to follow. 

It seems error bars are added , a table with statistics included and an additional paragraph in results.

Add a paragraph in materials & methods explaining your statistics and reason for choosing Kruskal wallis non parametric tests, and the software version used.

There is still two subtitles in results are the same (3.1).

Author Response

Thank you very much for your review and for your constructive comments!

The addition of comparative statistics to the manuscript took some time and consequently it was not possible to resubmit unde rthe original manuscript number but to do a new submission. The changes made as compared to previous versions were exactly as described i.e. adding comparative statistics. 

We have further elaborated on the tests used and the respective section now reads:

"The ratings obtained for mechanical debridement, air abrasion and 5 min BDD electrode treatment of implants (n = 3) covered by E. faecalis, C. dubliniensis and multi-species biofilm were subject to comparative statistical analysis. To this end, the sums of the individual ratings were used. In view of the small sample sizes, the integer valued results cannot be assumed to be normally distributed. Therefore, the parameter-free Kruskal Wallis rank sum test (R, version 3.6.2, The R Foundation for Statistical Computing, Vienna, Austria; www.R-project.org) was applied to compare the different treatments. The level of significance was set at α=0.05."

Based on your comment we have changed and double checked the numbering of subheadings

This manuscript is a resubmission of an earlier submission. The following is a list of the peer review reports and author responses from that submission.

Round 1

Reviewer 1 Report

The authors conducted a study aiming to test the application of boron-doped diamond (BDD) electrodes for the electrochemical disinfection of dental implants colonized by biofilm-forming microorganisms. The topic is clinically relevant but the defect model of the study may not be generalizable. I have the following comments related to the study design and outcomes:

Introduction

1. page 2: “While the exact patho-mechanisms seem not to be fully understood, it appears to be consensus that bacterial biofilms on dental implants [13,17] can cause an inflammatory reaction [12,18,19] resulting in loss of periimplant bone [20].” Please add citation from the 2017 World Workshop:

Peri-implant diseases and conditions: Consensus report of workgroup 4 of the 2017 World Workshop on the Classification of Periodontal and Peri-Implant Diseases and Conditions. J Clin Periodontol. 2018 Jun;45 Suppl 20:S286-S291.

2. page 4: “Mechanical debridement (curettes…)” Please specify the type of the curettes used. Is it plastic, titanium, graphite or stainless steel?

M&M

1. page 3: “The microorganisms applied included yeasts (Candida albicans, Candida dubliniensis), Gram-negative (Roseomonas mucosa) and Gram-positive bacteria (Enterococcus faecalis, Staphylococcus sanguinis, Streptococcus epidermidis)…” There is no periodontal pathogens (i.e. P. gingivalis, T. forsythia) used in your model. Since this is a peri-implantitis model, facultative gram-negative bacteria is more clinically relevant. Any reason not to use this model in vitro?

2. “polyurethane foam blocks mimicking alveolar bone…” Any literature to support this model since alveolar bone is not simply a foam blocks, it can be very complicated in different regions of maxillary/mandibular arches. Please refer to Lekholm and Zarb 1985 specifying Type I to Type IV bone density. Your foam model only represents a Type IV bone density.

3. “osteotomies were created in preformed saucer-shaped defects applying the regular surgical protocol…” Did you mimic the peri-implant bone defect? If yes, please specify the type of peri-implant bone defect based on Schwarz et al. 2010:

Schwarz F, Sahm N, Schwarz K, Becker J. Impact of defect configuration on the clinical outcome following surgical regenerative therapy of peri-implantitis.

Results

1. Page 7-8: I highly recommend to conduct a few more samples to show the efficiency of BDD with various types of peri-implant defect. Most of the peri-implant defects have a subcrestal component, which is extremely difficult to decontaminate. The subcrestal component of the defect can be easily generated with the bovine ribs. Without showing the efficiency in the subcrestal peri-implant defect, the true benefit of using BDD is doubtful.

Discussion

1.     Please specify the study limitation and future directions in the Discussion section.

2.     Was there any microscopic examination done to assess the outcome of the surface decontamination?

Author Response

The authors conducted a study aiming to test the application of boron-doped diamond (BDD) electrodes for the electrochemical disinfection of dental implants colonized by biofilm-forming microorganisms. The topic is clinically relevant but the defect model of the study may not be generalizable. I have the following comments related to the study design and outcomes:

Re: The reviewer is correct with respect to the periimplant defect simulated here which certainly cannot perfectly mimic clinical conditions. As already cited in the manuscript, we have applied the same defects as done by Wei et al. 2017 and as per your request, we have further specified the defects as “circular bone resorption under maintenance of the buccal and oral compacta”

Introduction

page 2: “While the exact patho-mechanisms seem not to be fully understood, it appears to be consensus that bacterial biofilms on dental implants [13,17] can cause an inflammatory reaction [12,18,19] resulting in loss of periimplant bone [20].” Please add citation from the 2017 World Workshop: Peri-implant diseases and conditions: Consensus report of workgroup 4 of the 2017 World Workshop on the Classification of Periodontal and Peri-Implant Diseases and Conditions. J Clin Periodontol. 2018 Jun;45 Suppl 20:S286-S291.

Re: Thank you very much for pointing us to this publication, which we have now added as a reference

page 4: “Mechanical debridement (curettes…)” Please specify the type of the curettes used. Is it plastic, titanium, graphite or stainless steel?

Re: We have added that stainless steel curettes have been used (EXD11/12, HuFriedy, Chicago, IL, USA)

M&M

page 3: “The microorganisms applied included yeasts (Candida albicans, Candida dubliniensis), Gram-negative (Roseomonas mucosa) and Gram-positive bacteria (Enterococcus faecalis, Staphylococcus sanguinis, Streptococcus epidermidis)…” There is no periodontal pathogens (i.e. P. gingivalis, T. forsythia) used in your model. Since this is a peri-implantitis model, facultative gram-negative bacteria is more clinically relevant. Any reason not to use this model in vitro?

Re: We are aware that the species mentioned by the reviewer are important members of dental and implant biofilms. However, these microorganisms are either anaerobic or microaerophilic and, consequently, even impaired by standard oxygen concentrations. For our experiments, we have chosen much more robust species. As suggested by the reviewer, this selection is better explained now.

“polyurethane foam blocks mimicking alveolar bone…” Any literature to support this model since alveolar bone is not simply a foam blocks, it can be very complicated in different regions of maxillary/mandibular arches. Please refer to Lekholm and Zarb 1985 specifying Type I to Type IV bone density. Your foam model only represents a Type IV bone density.

Re: The reviewer is correct in stating that human bone is way more complex than a simplistic polyurethane foam. Nevertheless, this material was used as it is well standardized and hence allowed for comparable conditions for the different treatment modalities applied here. We have added the Lekholm and Zarb classification and also added this point as a limitation of this investigation.

“osteotomies were created in preformed saucer-shaped defects applying the regular surgical protocol…” Did you mimic the peri-implant bone defect? If yes, please specify the type of peri-implant bone defect based on Schwarz et al. 2010: Schwarz F, Sahm N, Schwarz K, Becker J. Impact of defect configuration on the clinical outcome following surgical regenerative therapy of peri-implantitis.

Re: We have added that type Ie defects have been simulated - Please also see response to first comment

Results

Page 7-8: I highly recommend to conduct a few more samples to show the efficiency of BDD with various types of peri-implant defect. Most of the peri-implant defects have a subcrestal component, which is extremely difficult to decontaminate. The subcrestal component of the defect can be easily generated with the bovine ribs. Without showing the efficiency in the subcrestal peri-implant defect, the true benefit of using BDD is doubtful.

Re: We fully agree with the reviewer that subcrestal defects are critical and this was the reason why we tried to simulated this clinical scenario which we hopefully have now described sufficiently. Unfortunately, Fig. 3B and 3C require a very close look to identify the saucer-shaped defects created in the circumference of the implants.

Discussion

Please specify the study limitation and future directions in the Discussion section.

Re: We have further elaborated on the limitations this pilot investigation had

Was there any microscopic examination done to assess the outcome of the surface decontamination?

Re: The implant surfaces have been analysed under SEM and the findings have been reported in “Göltz, M., Koch, M., Detsch, R., Karl, M., Burkovski, A. & Rosiwal, S. (2019) Influence of in-situ electrochemical oxidation on implant surface and colonizing microorganisms evaluated by scatter electron microscopy. Materials 12, 3977.”

We have added these findings to the discussion section and referenced the paper.

Reviewer 2 Report

The paper referenced as #671623 by JCM titled as “Electrochemical Disinfection of Dental Implants Experimentally Contaminated with Microorganisms as a Model for Periimplantitis” seems to be a follow up of a previous paper by the authors, who used similar technology to disinfect extracted teeth. In the current paper authors used  a specific boron-doped diamond coated niobium wire electrode to disinfect previously biofilm colonized dental implant surfaces. 

The fundamental criticism to this research is the quantitative methods as reported in this paper.  Authors used 5 Straumann SLA surface design dental implants , but it is not clear if 5 implants actually used for each experimental group.  The actual colony/bacteria measurement method is not explained and only referenced to a previous article , which I believe has been misquoted in the text (quoted 47 , I believe it should be 46).  A brief summary of quantitation technique should be included in this paper.  There is no statistics used , and bar graphs have no error bar (SE) or standard deviation (SD).  Are we looking at only one sample? Or if there were 5 implants per group, please express error values and appropriate statistics to rule out if the findings simply occurred by chance.

There are two subtitle under “3.3. Time-Dependent Removal of Biofilm from Implants Contaminated with Different Microorganisms”  repeated in the results for different sections.

It is not clear how and where this electrode is going to be applied in a real clinical situation? I guess it does not contact the implant ?

Author Response

The paper referenced as #671623 by JCM titled as “Electrochemical Disinfection of Dental Implants Experimentally Contaminated with Microorganisms as a Model for Periimplantitis” seems to be a follow up of a previous paper by the authors, who used similar technology to disinfect extracted teeth. In the current paper authors used a specific boron-doped diamond coated niobium wire electrode to disinfect previously biofilm colonized dental implant surfaces.

Re: The reviewer is correct, we have applied the electrochemical disinfection apparatus both for endodontics and implantology and the paper has now been referenced

The fundamental criticism to this research is the quantitative methods as reported in this paper. Authors used 5 Straumann SLA surface design dental implants , but it is not clear if 5 implants actually used for each experimental group.

Re: The treatment protocol is better explained now (l. 136-137): For every species investigated at least 3 biological replicates were tested for each treatment procedure.

The actual colony/bacteria measurement method is not explained and only referenced to a previous article , which I believe has been misquoted in the text (quoted 47 , I believe it should be 46). A brief summary of quantitation technique should be included in this paper.

Re.: We extended the explanation of the method and added a new reference (l. 140-142)

There is no statistics used , and bar graphs have no error bar (SE) or standard deviation (SD).  Are we looking at only one sample? Or if there were 5 implants per group, please express error values and appropriate statistics to rule out if the findings simply occurred by chance.

Re: Done, see figure 5 to 7 and corresponding legends.

There are two subtitle under “3.3. Time-Dependent Removal of Biofilm from Implants Contaminated with Different Microorganisms”  repeated in the results for different sections.

Re: Sorry for the mistake. 2nd title changed to “3.4. Influence of BDD Electrode Treatment on Temperature and pH”

It is not clear how and where this electrode is going to be applied in a real clinical situation? I guess it does not contact the implant ?

Re: We are aware that the device is far away from clinical application requiring several development steps. While GalvoSurge uses the implant itself as an electrode, the apparatus envisaged here is a separate probe contained in e.g. a permeable plastic tube in order not to touch a metallic implant. We have tried to include a vague perspective in the discussion but are hesitant at the current stage to propose further details.

Round 2

Reviewer 1 Report

The authors have addressed the issues I raised. Therefore I recommend acceptance for publication.

Author Response

The authors have addressed the issues I raised. Therefore I recommend acceptance for publication.

Re: Thank you

Reviewer 2 Report

The authors addition improved the paper but it still did not address some of my criticisms :

1- What statistical tests were used , Where is the p value , where is the details of statistical tables.

2- Were data normally distributed?

3-  Design  a table that shows how these 45 implants used.  How were these implants distributed? Give details of sample size per group.

4- I am not sure , why authors claim they are deliberately being vague in explaining a methodology in a scientific journal?  I understand this method is the subject of a patent , and I don't believe publishing details will risk the patent since it is already filed or is in the process of filling. Detailed methodology must be published in scientific peer-reviewed journals ; otherwise , it can not be properly reviewed or its repeatability verified.

Author Response

The authors addition improved the paper but it still did not address some of my criticisms :

1- What statistical tests were used, Where is the p value , where is the details of statistical tables.

2- Were data normally distributed?

Re: Both, reviewer remarks #1 and #2 would require comparative statistical analysis, which was not done here. As now stated more explicitly at the end of the introduction section, this was a preliminary proof of principle study i.e. we wanted to show that the BDD electrode is capable of removing biofilm at all. The control treatments were primarily performed as a test for our workflow. We have added this aspect also to the discussion section.

3-  Design  a table that shows how these 45 implants used.  How were these implants distributed? Give details of sample size per group.

Re: We have added supplementary table 1 detailing the exact numbers and we have added to the materials and methods section that the implants could be reused following sterilization

4- I am not sure , why authors claim they are deliberately being vague in explaining a methodology in a scientific journal?  I understand this method is the subject of a patent , and I don't believe publishing details will risk the patent since it is already filed or is in the process of filling. Detailed methodology must be published in scientific peer-reviewed journals ; otherwise , it can not be properly reviewed or its repeatability verified.

Re: It was not our goal to hide relevant information regarding the BDD electrodes. The basic setup of the device in our opinion is well described and shown in this manuscript and we have referenced a recent paper even detailing the BDD coating process (Reference #54). We honestly present the current state of development but are certainly willing to share any specific detail the reviewer requests.